# A diel multi-tissue genome-scale metabolic model of *Vitis vinifera*

**Marta Sampaio**[1]*, **Miguel Rocha**[1,2], **Oscar Dias**[1,2]*

**1** Centre of Biological Engineering, University of Minho, Campus of Gualtar, Braga, Portugal, **2** LABBELS, Associate Laboratory, Braga/Guimarães, Portugal

* msampaio@ceb.uminho.pt; odias@ceb.uminho.pt

**Data Availability Statement:** All the raw data required to replicate the results of our study are available in the supplementary material and the metabolic models created in this work are available in the BioModels database.

## Abstract

*Vitis vinifera*, also known as grapevine, is widely cultivated and commercialized, particularly to produce wine. As wine quality is directly linked to fruit quality, studying grapevine metabolism is important to understand the processes underlying grape composition. Genome-scale metabolic models (GSMMs) have been used for the study of plant metabolism and advances have been made, allowing the integration of omics datasets with GSMMs. On the other hand, Machine learning (ML) has been used to analyze and integrate omics data, and while the combination of ML with GSMMs has shown promising results, it is still scarcely used to study plants. Here, the first GSSM of *V. vinifera* was reconstructed and validated, comprising 7199 genes, 5399 reactions, and 5141 metabolites across 8 compartments. Tissue-specific models for the stem, leaf, and berry of the Cabernet Sauvignon cultivar were generated from the original model, through the integration of RNA-Seq data. These models have been merged into diel multi-tissue models to study the interactions between tissues at light and dark phases. The potential of combining ML with GSMMs was explored by using ML to analyze the fluxomics data generated by green and mature grape GSMMs and provide insights regarding the metabolism of grapes at different developmental stages. Therefore, the models developed in this work are useful tools to explore different aspects of grapevine metabolism and understand the factors influencing grape quality.

## Author summary

Grapevine is one of the most important crops in the world, mainly due to wine production. Understanding grape metabolism is key to improving fruit quality. Towards this aim, we reconstructed a generic genome-scale metabolic model of grapevine to represent grapevine metabolism *in silico* as well as tissue-specific models by integrating omics data of stem, leaf, and berry. These models were combined into day-night multi-tissue models to explore grapevine metabolism under different conditions. Then, we trained machine learning models with the fluxomics data predicted by the metabolic models to identify the changes in grape metabolism during development. This first effort of integrating omics, metabolic models and machine learning showcases the potential of applying this strategy for more insightful analyses, as additional data becomes available.

**Funding:** Portuguese Foundation for Science and Technology (FCT) supported this work under the scope of the strategic funding of UID/BIO/04469/2020 unit and the PhD scholarship (SFRH/BD/144643/2019) to M.S. FCT also celebrated an Assistant Research contract with O.D. obtained under CEEC Individual 2018 (DOI 10.54499/CEECIND/03425/2018/CP1581/CT0020). The funders had no role in study design, data collection and analysis, decision to publish, or preparation of the manuscript.

**Competing interests:** The authors have declared that no competing interests exist.

## Introduction

*Vitis vinifera* is one of the major fruit crops in the world. It is cultivated worldwide and has high economic value, mainly due to wine production. In 2022 and despite inflation, wine exports reached a value of 37.6 billion euros [1]. In addition, grapes have other purposes, being marketed as fresh and dried fruits, and used for juice production. The grape pulp contains high levels of sugars and phenolic compounds, like flavonoids and stilbenes, with potential health benefits, such as antioxidant and anti-inflammatory activities, and cardiovascular protection [2], thus currently being studied for possible pharmaceutical and cosmetic applications. Therefore, as grapevines have high economic interest and fruit quality is intrinsically linked to metabolism, the study of grapevine metabolism is essential for understanding its responses to different environmental conditions that may affect grape metabolic composition.

Genome-scale metabolic models (GSMMs) represent all metabolic reactions taking place within an organism. These models are reconstructed from the genome and allow performing phenotype predictions under different environmental or genetic conditions [3]. Although GSMMs have been extensively used for the metabolic engineering of prokaryotes, several GSMMs are available for plants [4], mainly *Arabidopsis thaliana* [5–11], *Zea mays* [7,12,13], and *Oryza sativa* [14–16]. Currently, the reconstruction of plant GSMMs is still very challenging and time-consuming due to the high number of gaps in genome annotations, the large diversity of metabolites, and the extensive compartmentalization of plant cells [17–19]. Despite the obstacles, many plant GSMMs have emerged recently, and new approaches have been developed to reconstruct more realistic models that include different plant tissues, through the integration of omics data [20,21], as well as the day-night cycle [9,22,23]. These models allow for differentiating the metabolism of each tissue and analyzing the metabolic interactions between tissues and the light and dark phases.

Despite the existence of several methods for integrating omics into GSMMs, this is still a challenging and inefficient task. As omics datasets are complex and heterogeneous, Machine Learning (ML) has been used to process and integrate different types of omics to extract biological knowledge from data. Recently, ML and GSMM approaches have been combined to improve the model's predictions and interpretability, and this strategy has shown promising results [19,24–27]. ML can be used to extract knowledge from the fluxomics data generated by the models or to integrate the predicted fluxomics data with experimental omics. Thus far, these studies have mainly been applied to bacteria, yeast, and human cells, but not to plants.

In this manuscript, we pioneer *V. vinifera* research with the introduction of iMS7199, the first GSMM for the grapevine, developed using the most recent genome version, PN40024.v4 [28]. In addition to the overarching model, tissue-specific models for the leaf, stem, and grape were developed by incorporating RNA-Seq data from these distinct tissues. Furthermore, to capture the changes in grape metabolism, we created two separate models representing the grape in both its green and mature states. These tissue-specific models were then integrated to construct diel multi-tissue GSMMs, enabling the simulation of grapevine metabolism across the day-night cycle and facilitating the study of inter-tissue metabolic interactions. Utilizing this comprehensive model, we investigated the metabolic responses of the grapevine under varying concentrations of sulfate and nitrate.

Also, simulated fluxomics data were generated from GSMMs of grapes in the green and mature state and analyzed by ML to identify the reactions that most contribute to the model's predictions of the grape developmental phase.

Therefore, this diel multi-tissue GSMM emerges as a useful tool for exploring the metabolic behaviour of *V. vinifera* under various conditions, offering insights into factors influencing

grape quality and phenolic content. In addition, the analysis of generated data from GSMMs by ML represents the first effort to apply this strategy in the study of plant metabolism.

## Results and discussion

### The *iplants* repository

To collect and organize all the relevant data for the model reconstruction efforts, a repository with the metabolic information of PlantCyc 14.0 [29] and MetaCyc 26.1 [30] databases, and Universal Protein Resource (UniProt) [31] sequence data was created. In total, the repository includes 24333 metabolites, 20518 reactions, 3519 pathways, and 22433 enzymes, 72% of which have a protein sequence. Details on how data is organized in the *iplants* repository are available in S2 File.

In addition to data from the metabolic databases, nine plant metabolic models were integrated into the *iplants* repository, namely *Arabidopsis thaliana* [5,8], *Zea mays* [13], *Oryza sativa* [14,16], *Solanum lycopersicum* [32], *Medicago truncatula* [33], *Glycine max* [34], and *Setaria viridis* [35]. These models have PlantCyc and MetaCyc identifiers for metabolites and reactions, which facilitated the integration. In total, 3815 metabolites and 4197 reactions from the models were successfully integrated.

*iplants* repository can be accessed through an application programming interface (API) created with Django and Django REST framework for both database systems, using Mongoengine and Neomodel Python packages. Several views were defined to allow the extraction of the data needed for the reconstruction of GSMMs and to save the data of the model under reconstruction (https://iplantsdb.bio.di.uminho.pt/).

### Model properties

A GSMM for *V. vinifera* was reconstructed from the PN40024.v4 genome (annotation version 1) [28]. DIAMOND similarity searches [36] against *iplants* resulted in 10840 protein matches, representing 26% of the 41160 proteins in the genome, which is in line with the percentage of metabolic genes described for the *A. thaliana*'s genome (between 25–30%) [37].

The reconstructed generic model, iMS7199, includes 5399 reactions (1624 transporters and 244 exchanges), and 5141 metabolites, across eight compartments: cytosol, chloroplast, mitochondria, endoplasmic reticulum, peroxisome, Golgi apparatus, vacuole, and extracellular space. In this model, the Gene-Protein-Reaction (GPR) rules were defined using the genome protein identifiers instead of genes as genome annotation was performed using protein sequences. As genes can encode more than one protein, the model includes 7199 protein identifiers that represent the 6018 genes of the *V. vinifera* genome.

This model is mass-balanced and can simulate growth in phototrophic and heterotrophic conditions, by setting the photon and carbon dioxide or sucrose as the only energy or carbon source, respectively. In addition, it requires the uptake of nitrate, phosphate, sulfate, iron, magnesium, and water to produce biomass.

The statistics of the *V. vinifera* model, as well as other relevant plant models, are presented in Table 1. Analyzing the table, only the *Quercus suber* model [38] has more genes, reactions, and metabolites than the *V. vinifera* model. The other models are much smaller, even the *G. max* model, which has a high number of genes.

The reactions of *V. vinifera* were compared with those from the other models, except for *Q. suber* which has different model identifiers. Drains, transporters, biomass pseudo-reactions, and compartments were not considered, resulting in 2769 reactions of the iMS7199 model (Fig 1).

**Table 1. Statistics of the *V. vinifera* model and other eight plant GSMMs.**

| | Reactions | Metabolites | Genes | Compartments |
|---|---|---|---|---|
| *A. thaliana* (Cheung et al. [8]) | 2769 | 2739 | 2857 | 5 |
| *Z. mays* (Bogart et al. [13]) | 1268 | 1121 | 2140 | 8 |
| *O. sativa* (Chatterjee et al. [16]) | 1136 | 1330 | 3602 | 4 |
| *S. lycopersicum* (Yuan et al. [32]) | 2143 | 1998 | 3410 | 5 |
| *M. truncatula* (Pfau et al., [33]) | 2909 | 2780 | 3403 | 8 |
| *G. max* (Moreira et al., [34]) | 3001 | 2814 | 6127 | 5 |
| *S. viridis* (Shaw et al. [35]) | 2473 | 2429 | 3376 | 5 |
| *Q. suber* (Cunha et al. [38]) | 6230 | 6481 | 7871 | 8 |
| *V. vinifera* (this work) | 5399 | 5141 | 7199 | 8 |

The *G. max* model shares the highest number of reactions with the *V. vinifera* model, corresponding to around 41% of all reactions from both models. This model is followed by the ones of *S. viridis* and *A. thaliana*, which share 1326 (38%) and 1342 (36%) reactions with iMS7199, respectively. The most distant model is the one from *O. sativa*, sharing only 624 reactions (20%).

In total, *V. vinifera* has 785 reactions that are not present in any other model (S3 Fig in S1 File). These reactions were analyzed to identify the associated pathways and gene annotation. Reactions without pathway associations were not considered.

The biosynthesis of secondary metabolites is the pathway class associated with more unique reactions, around 140, followed by fatty acid biosynthesis (88 reactions), protein glycosylation (35 reactions), and fatty acid and lipid degradation (17 reactions). These pathway classes comprise several specific pathways. Other specific pathways with more than four unique reactions include cholesterol biosynthesis and diacylsucrose biosynthesis. Hence, the *V. vinifera* model represents a great advance compared to the previous plant models, as it comprises new reactions, especially for the secondary metabolism, which is often underrepresented in plant models.

**REACTON CONTENT**

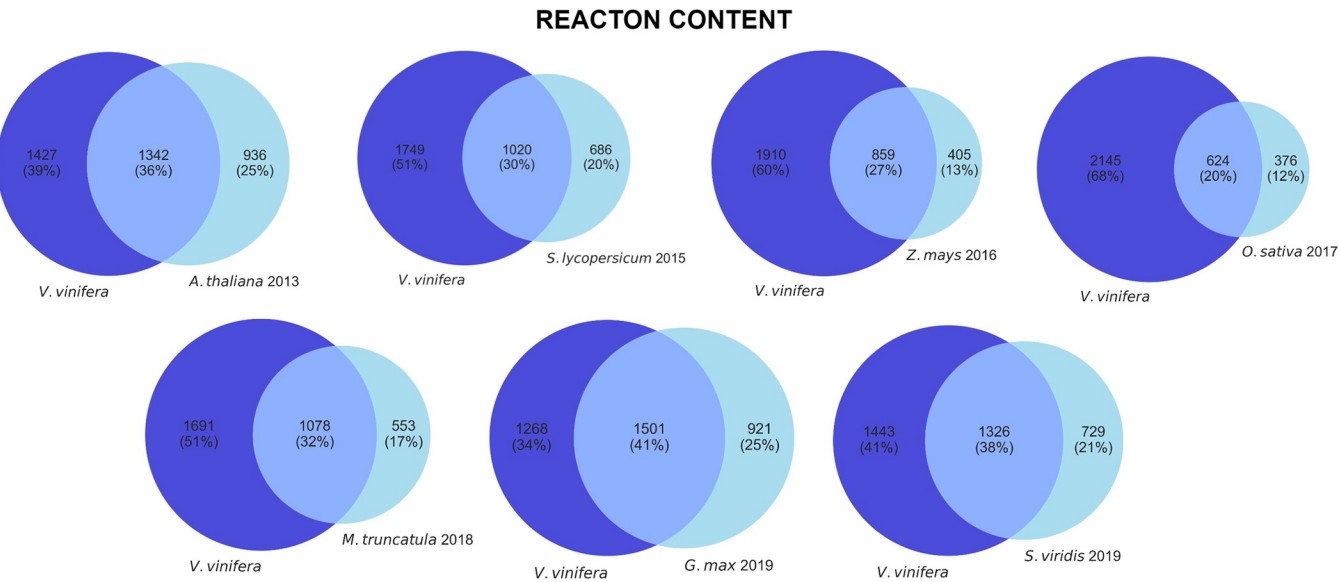

**Fig 1. Venn diagrams comparing the reaction content of the *V. vinifera* model with the other seven plant models.**

On the other hand, the other plant models include 661 secondary pathway reactions not available in iMS7199. This number can be explained by the fact that 380 reactions are not associated with GPR rules in the models. Moreover, 93 are sub-reactions of others that are included in the *V. vinifera* model. Of the 188 reactions that have GPR rules, 155 have a corresponding enzyme sequence in the *iplants* database, which had a match in the DIAMOND annotation but were not the first hit for any query protein (S3 File). This data is made available and may be used in the future to improve the model by performing further manual curation. Regarding genome annotation, 27% of the proteins that catalyze these unique reactions were annotated based on the genome of *A. thaliana*. These proteins or reactions were probably not in metabolic databases when the *A. thaliana* models were reconstructed, which can explain why they are missing from these models. Besides *A. thaliana*, 12% of the unique proteins matched human proteins, and around 27% were annotated based on proteins from more than 100 different plant species, including *S. lycopersicum*, *Solanum tuberosum*, *Catharanthus roseus*, *Petunia x hybrida*, *M. truncatula*, *G. max*, and *V. vinifera*.

As *A. thaliana* is a reference organism for plants, there are several GSMMs for this organism [5–11] and much enzymatic and metabolic information about *A. thaliana* is available in databases. On the other hand, data for more complex plants is scarce. Therefore, it was expected that a large percentage of *V. vinifera* proteins would be annotated based on homologous proteins from *A. thaliana*. However, as *V. vinifera* is a much more complex plant, gene annotations can be wrong or missing, and the consequent validation process helps to limit these errors. In addition, several proteins were similar to human proteins, which was also expected as various pathways, mainly related to lipid metabolism, are better characterized in humans than in plants.

## Tissue-specific models

Tissue-specific models were reconstructed to represent the metabolic differences between tissues. This was accomplished by integrating RNA-Seq data with the iMS7199 model.

**RNA-Seq Data.** The RNA-Seq data of *V. vinifera* Cabernet Sauvignon was retrieved from the GREAT database [39] for leaf, stem, and berry. In total, the RNA-Seq dataset contained the expression of the 6018 genes (matching the 7199 proteins in the model) across 162 samples. The time-point metadata for berry samples was discretized into two developmental stages, green and mature. Mature berry is the most represented tissue in the dataset with 46% of the samples (75 samples), 28% of the samples are from green berries (45 samples), while stem and leaf represent 13% of the samples each (21 samples).

**Biomass composition.** The biomass composition of the different tissues is represented in Fig 2. Details on biomass compositions are described in the Materials and Methods section and S4 File. The biomass composition of leaf and green berry was considered equal and it was used as a reference to define the biomass composition of the other tissues. According to other plant models (*Q. suber* [38] and *A. thaliana* [10]), the stem is expected to have a higher cell wall and carbohydrate content and lower protein and lipid levels. According to the literature, the mature berry is expected to have higher sugar and amino acid content [40]. Therefore, in the model, leaf and green berries present high levels of carbohydrates and proteins, the stem is mainly composed of carbohydrates and cell wall precursors, and mature berries present high amounts of sugars and proteins but fewer organic acids.

**Models.** The FASTCORE algorithm [41] was used to create tissue-specific models (see Materials and Methods). The statistics of the reconstructed generic and tissue-specific models are shown in Table 2. All models have the same 244 exchange reactions.

The number of reactions is similar across all tissue-specific models. Even so, the mature berry model is the smallest one, while the leaf model is the largest, having a higher number of

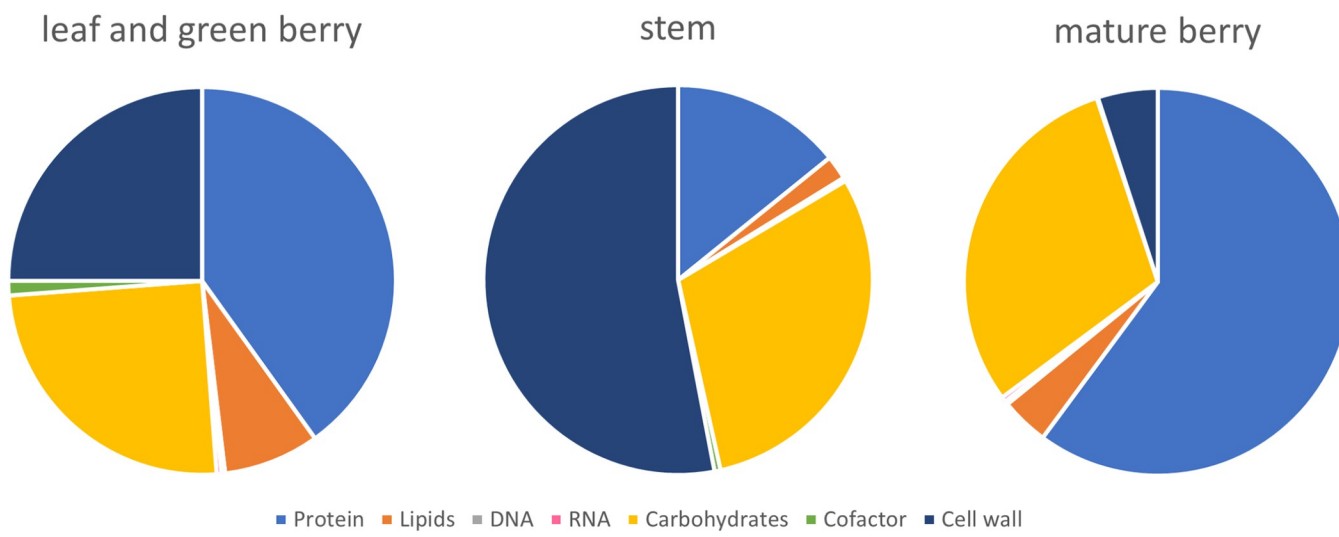

**Fig 2. Biomass percentage for the leaf and green berry, stem, and mature berry.** These values were adapted from available plant models and literature.

reactions in the chloroplast, as well as more unique reactions. At the pathway level, no significant differences were found between models (S5 File).

The flux distributions of the tissue-specific models were simulated using parsimonious Flux Balance Analysis (pFBA) [42], following the first strategy defined in the Materials and Methods section of keeping the biomass rate at 0.11 h$^{-1}$ and minimizing the uptake of photons and sucrose.

A summary of the phenotype predictions is presented in Table 3 and full results are available in S6 File. The flux distribution in the leaf tissue was simulated for all processes: photosynthesis, photorespiration, and respiration, while in the other tissues it was simulated for respiration only. Photosynthesis can also occur in green berries, but not at significant levels [43]. Hence, in this analysis, the leaf was considered to be the only photosynthetic tissue.

In photosynthesis and photorespiration, the leaf uptakes light, carbon dioxide, water, nitrate, sulfate, and protons to produce biomass, and releases oxygen and phosphate, as

**Table 2. Statistics of the generic and tissue-specific GSMMs of *V. vinifera*.**

|  | Generic model | Leaf | Stem | Green berry | Mature berry |
|---|---|---|---|---|---|
| Genes | 7199 | 6701 | 6602 | 6657 | 6312 |
| Metabolites | 5141 | 4456 | 4310 | 4399 | 4181 |
| Reactions | 5399 | 4510 | 4384 | 4495 | 4272 |
| Transport | 1624 | 1295 | 1315 | 1324 | 1305 |
| Unique reactions | - | 124 | 97 | 26 | 19 |
| Metabolic reactions | 3531 | 2971 | 2825 | 2927 | 2723 |
| Cytosol | 1434 | 1154 | 1092 | 1113 | 1059 |
| Chloroplast | 793 | 745 | 701 | 725 | 684 |
| Mitochondria | 335 | 313 | 318 | 320 | 314 |
| Endoplasmic reticulum | 568 | 410 | 370 | 417 | 353 |
| Peroxisome | 165 | 158 | 152 | 153 | 151 |
| Vacuole | 52 | 49 | 47 | 48 | 32 |
| Golgi complex | 54 | 41 | 41 | 50 | 50 |
| Extracellular | 130 | 101 | 104 | 101 | 80 |

**Table 3. Summary of the net conversions obtained from the phenotype predictions of the tissue-specific models of leaf, stem, green berry, and mature berry for photosynthesis, photorespiration (leaf only), and respiration, minimizing the uptake of photons or sucrose and fixing biomass rate at 0.11h⁻¹.** This table shows the metabolites that are consumed and produced by the models. The fluxes of the metabolites are in mmol.gDW⁻¹.h⁻¹ while biomass fluxes are in h⁻¹.

| metabolite | photosynthesis | photorespiration | respiration | | |
| --- | --- | --- | --- | --- | --- |
| | leaf | leaf | stem | berry green | berry mature |
| **Uptake** | | | | | |
| SUCROSE | - | - | 0.61 | 0.49 | 0.61 | 0.72 |
| Light | 32.09 | 43.76 | - | - | - | - |
| CARBON-DIOXIDE | 4.41 | 4.41 | - | - | - | - |
| NITRATE | 0.37 | 0.37 | 0.37 | - | 0.37 | - |
| OXYGEN-MOLECULE | - | - | 1.74 | 1.71 | 1.74 | 1.84 |
| PROTON | 6.64 | 6.64 | 3.73 | 4.01 | 3.73 | 3.71 |
| SULFATE | 0.02 | 0.02 | 0.02 | 0.01 | 0.02 | 0.02 |
| WATER | 3.21 | 3.21 | - | 0.11 | - | - |
| **Production** | | | | | |
| OXYGEN-MOLECULE | 5.54 | 5.54 | - | - | - | - |
| NITRATE | - | - | - | - | - | 0.70 |
| AMMONIUM | - | - | - | 0.04 | - | - |
| HCO3 | - | - | 2.86 | 1.98 | 2.86 | 3.47 |
| PPI | 0.05 | 0.05 | - | - | - | - |
| Pi | - | - | 0.09 | 0.21 | 0.09 | - |
| WATER | - | - | 0.55 | - | 0.55 | 1.25 |
| e-Biomass | 0.11 | 0.11 | 0.11 | 0.11 | 0.11 | 0.11 |

expected. Iron II and magnesium (Mg) are also captured but with very low fluxes (less than 1e-5 mmol.gDW⁻¹.h⁻¹). Light uptake is significantly higher in photorespiration than in photosynthesis (43.75 mmol.gDW⁻¹.h⁻¹ *vs* 32.09 mmol.gDW⁻¹.h⁻¹), which was also expected as the latter process is known to be more efficient for energy production. In both cases, the pathways of the primary metabolism are the most active: photosynthesis, Calvin cycle, glycolysis, starch and amino acid biosynthesis, and oxidative phosphorylation. Also, the photorespiration pathway is only active under photorespiration conditions.

During photosynthetic conditions, the tricarboxylic acid (TCA) cycle is incomplete: citrate is converted to isocitrate, and this is converted to α-ketoglutarate, which is used for the biosynthesis of glutamate and glutamine instead of being used to produce succinate. Fumarate is produced from arginine biosynthesis, instead of being produced from succinate, and enters the cycle. This result is consistent with the results observed for other plant models under light conditions [9,38] and with isotope labelling experiments, which stated that a cyclic TCA only happens when the demand for ATP is high. The photosynthetic ATP production reduces that demand [44,45].

In respiration, the leaf uptakes sucrose, nitrate, sulfate, oxygen, and protons, and releases hydrogencarbonate, water, and phosphate. The main active pathways include glycolysis, the TCA cycle, starch and amino acid biosynthesis, and oxidative phosphorylation. The respiration results were similar across tissues. The stem uptakes water and releases ammonium, and the mature berry has a slightly higher demand for sucrose to produce the same biomass flux.

In summary, the integration of omics data into the generic GSMM created tissue-specific models that try to reflect the differences in gene expression among models. However, the number of reactions and the phenotype predictions are not very different between models; thus, a complementary analysis based on differential flux predictions was performed to understand the metabolic differences between the tissues.

**Differential flux analysis.** The ACHR sampler [46] was used to generate 10000 sample fluxes for all reactions from the different tissue-specific models. Then, these data were used to identify the reactions with differential fluxes between models (see Materials and Methods). In total, 764 reactions were found to have altered fluxes between at least two models.

Hypergeometric enrichment tests were used to identify the pathways that presented significantly differential flux between each pair of models. These results are available in the S7 File. Analyzing the results, it was clear that smaller pathways were not selected even when only one reaction was not identified as having differential flux. Therefore, this method seems to be more suitable for analyzing pathways with a large number of reactions. For this reason, the complete list of reactions with differential flux between the models was also analyzed.

Comparing the green and mature berry models, reactions from glycolysis, TCA cycle, and nucleotide biosynthesis were identified as having differential flux. In addition, anthocyanin and quercetin biosynthesis exhibited more flux in the mature berry. This was expected as the mature berry has anthocyanins and a higher content of sugars in its biomass composition while demanding a lower content of nucleotides.

Comparisons between the other models are available in S1 File. In summary, it was expected that the primary metabolic pathways would be identified as having differential flux between tissues, as tissue models have different demands for biomass precursors, and produce energy by different processes: the leaf performs photosynthesis, while the others perform aerobic respiration. Besides these, no relevant pathways were found to characterize the specific metabolism of each tissue.

## Diel multi-tissue models

Diel multi-tissue models were created to analyze the metabolic interactions between the leaf, stem, and berry of *V. vinifera* in the light (day) and dark (night) phases of a diel cycle. Two models were created, one using the green berry tissue and the other using the mature berry. The resulting diel multi-tissue models include 32391 and 31999 reactions, and 29064 and 28710 metabolites for green and mature berries, respectively. The structure of the multi-tissue diel models is schematized in Fig 3, showing the different tissues, the diel phases, and the connections between them.

pFBA was used to simulate the flux distribution using the models, as described in the Materials and Methods section for photorespiration conditions. The phenotype predictions are available in the S8 File. A summary of the results is presented in Table 4 and the fluxes for the storage metabolites between light and dark phases are shown in Tables 5 and S1 in S1 File, for green and mature berries, respectively.

Significant differences were found between the light and dark phases, mainly in the leaf, as photosynthesis and photorespiration occur in this tissue. The light phase starts with photosynthesis light reactions and carbon dioxide fixation through the Calvin cycle in the leaf. The resulting carbohydrates are then used to produce all biomass precursors. Starch, sucrose, malate, and some amino acids are stored to be used in the leaf during the dark phase. At night, the active pathways include aerobic respiration, starch degradation, glycolysis, pentose phosphate, and citrate biosynthesis through the TCA cycle. Sucrose was expected to be produced at night but instead, the model uses fructose 6-phosphate from starch degradation to start glycolysis. The sucrose requirements for biomass are fulfilled by accumulating very small quantities of sucrose between light and dark phases (flux value less than 0.02 mmol.gDW$^{-1}$.h$^{-1}$). This is an artefact as the model finds sucrose transport to the dark phase less costly than producing it. However, sucrose production at night can be assured by forcing flux in the respective reactions. In addition, starch is the main carbon compound stored in the light, and it is degraded

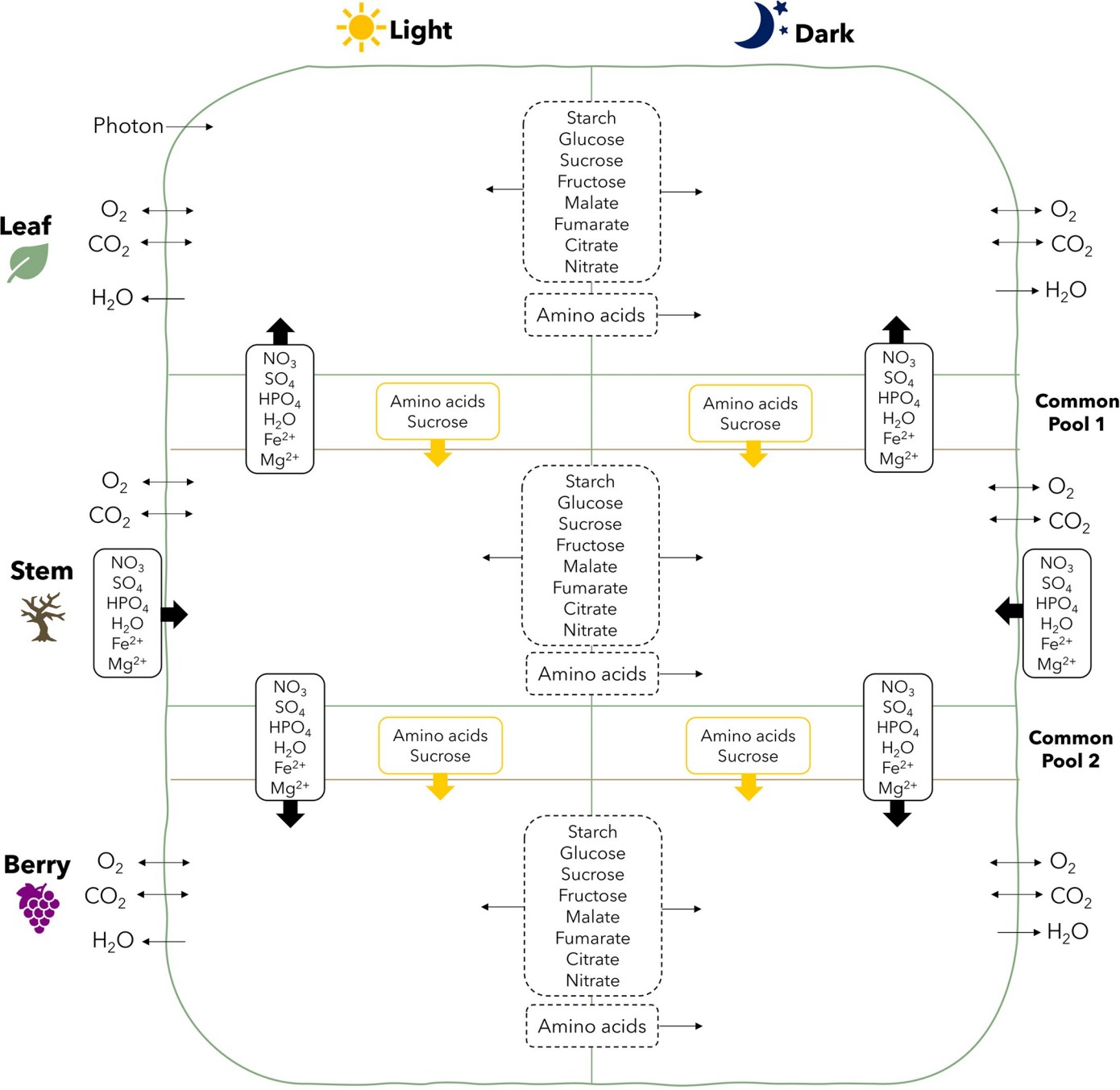

**Fig 3. Schematic representation of the reconstructed diel multi-tissue models of *V. vinifera*, including the leaf, stem, and berry tissues and the common pools 1 and 2 in both light and dark phases.** Photon uptake was allowed through the leaf in the light phase while mineral nutrients (nitrate, sulfate, phosphate, iron, magnesium) were allowed through the stem in both phases. Exchanges of carbon dioxide, oxygen, and water were allowed in all tissues and phases. Starch, glucose, sucrose, fructose, malate, fumarate, citrate, and nitrate were allowed to accumulate in the light and dark phases (dashed rectangle between phases). Amino acids can be stored in the light and used in the dark. Exchanges of amino acids, sucrose, and minerals were allowed between tissues through common pools.

in the dark phase to produce energy. This was expected as less energy is needed to mobilize plastidic starch reserves than vacuolar sucrose [9].

Then, the citrate produced at night is stored in the vacuole to be used during the day, entering the TCA cycle. Nitrate is also transported from the dark to the light to support nitrogen

**Table 4. Summary of the phenotype predictions for the diel multi-tissue models of green and mature berries under photorespiration with biomass maximization as objective function and fixing the photon uptake to 300 mmol.gDW$^{-1}$.h$^{-1}$.** This table shows the metabolites that are consumed and produced by the models. The fluxes of the metabolites are in mmol.gDW$^{-1}$.h$^{-1}$ while biomass fluxes are in h$^{-1}$.

| | photorespiration | |
|---|---|---|
| metabolite | green | mature |
| **uptake** | | |
| Light__light | 300.000 | 300.000 |
| NITRATE__light | 1.147 | 1.589 |
| NITRATE__dark | 0.764 | 1.059 |
| OXYGEN-MOLECULE_dark | 5.999 | 5.818 |
| PROTON_light | 26.330 | 27.210 |
| PROTON_dark | 16.790 | 16.020 |
| SULFATE_light | 0.098 | 0.112 |
| SULFATE_dark | 0.002 | 0.001 |
| WATER_light | 31.070 | 29.770 |
| CARBON-DIOXIDE_light | 35.830 | 34.920 |
| **production** | | |
| OXYGEN-MOLECULE_light | 39.810 | 40.84 |
| WATER_dark | 1.455 | 2.788 |
| CARBON-DIOXIDE_dark | 0.000 | 0.000 |
| HCO3_light | 2.995 | 2.956 |
| HCO3_dark | 5.602 | 4.475 |
| Pi_light | 0.527 | 0.384 |
| Pi_dark | 0.527 | 0.384 |
| total biomass | 0.149 | 0.142 |

assimilation, which was predicted to occur only during the day. These results were confirmed by experimental evidence and observed in other plant models [9,38,47]

However, the entire TCA cycle was expected to occur in the dark phase. This does not happen in the leaf, as all citrate produced at night is stored in vacuoles to be used during the day. α-ketoglutarate is produced from the degradation of amino acids like glutamate and enters the cycle, which is complete until citrate production. The citrate accumulated, besides feeding the TCA cycle in the light phase, is used for the biosynthesis of Acetyl Co-A during the day, which is then used for lipid production. Therefore, the model finds it more efficient to store more citrate to be used during the day than to complete the TCA cycle in the leaf at night.

Ammonium, phosphate, sulfate, pyruvate, formate, and glutamate are provided by the stem and transported to the leaf, where are used for amino acid and citrate biosynthesis.

On the other hand, sugars and amino acids produced in the leaf are transported to the stem. The active pathways in the stem during the day have much lower fluxes than in the leaf. These include sugar primary metabolism, amino acid and nucleotide biosynthesis, and degradation of beta-alanine and uracil.

The leaf and stem metabolisms in the dark phase are similar, and the same metabolites are stored between the light and dark phases. However, in the stem, the TCA cycle is complete during the night, as expected, but a high percentage of the produced citrate is still stored (around 59%). The berry metabolism is very similar to the stem metabolism, but the reaction fluxes are even lower, except for the reactions related to folate biosynthesis. Formate and pyruvate are produced here and transported to the stem through common pool 2 to be further transported from the stem to the leaf to be used for amino acid biosynthesis. Only starch and amino acids are exchanged in the berry between light and dark phases.

**Table 5. Fluxes for the metabolites stored between light and dark phases in the diel multi-tissue model with green berry.** Positive fluxes indicate that the metabolites are stored in the light phase to be used in the dark while the metabolites with negative fluxes are stored in the dark to be used during the day. The fluxes are in mmol.gDW$^{-1}$.h$^{-1}$.

|  | reaction | flux |
|---|---|---|
| leaf | **Citrate (CIT) storage** | **-1.354** |
|  | Cysteine (CYS) storage | 0.009 |
|  | Isoleucine (ILE) storage | 0.025 |
|  | Malate (MAL storage | 1.168 |
|  | Methionine (MET) storage | 0.011 |
|  | **Nitrate (NITRATE) storage** | **-0.020** |
|  | Proline (PRO) storage | 0.360 |
|  | Starch Storage | 0.374 |
|  | Sucrose storage | 0.020 |
|  | Threonine (THR) storage | 0.051 |
| stem | **Citrate (CIT) storage** | **-0.261** |
|  | Cysteine (CYS) storage | 0.003 |
|  | Isoleucine (ILE) storage | 0.009 |
|  | Malate (MAL storage | 0.060 |
|  | Methionine (MET) storage | 0.015 |
|  | **Nitrate (NITRATE) storage** | **-0.745** |
|  | Proline (PRO) storage | 0.276 |
|  | Starch Storage | 0.009 |
| berry | Cysteine (CYS) storage | 0.009 |
|  | Isoleucine (ILE) storage | 0.025 |
|  | Proline (PRO) storage | 0.162 |
|  | Starch Storage | 0.013 |

No significant differences were found in the phenotype predictions between green and mature berries. The total biomass rate is slightly higher in the green berry model (0.149 h$^{-1}$) than in the mature one (0.142 h$^{-1}$), and generally, the photosynthetic pathways and those related to cellular respiration have lower flux in the mature berry. The pathways related to secondary metabolite biosynthesis, mainly anthocyanins, have flux in the mature and not in the green berry, as expected, but these fluxes are very low; thus, no major differences were observed in the primary metabolism.

**Sulfate assimilation.** Sulfur is an important nutrient taken up by plants from the soil in the form of sulfate, and it is the key element of the amino acids cysteine and methionine. Sulfur is also a component of glutathione, which is an important antioxidant agent, and S-adenosyl methionine and coenzyme A, which are cofactors for several enzymes. Elemental sulfur (S$^0$) is the oldest pesticide applied to grapevines and it is still widely used nowadays, being particularly effective against powdery mildew disease, one of the most common diseases affecting grapevines that is caused by the fungus *Erysiphe necator*. In addition, sulfur dioxide (SO$_2$) is often used as a conservative of table grapes or in winemaking to prevent oxidation and microbial contamination.

Plant exposure to high sulfur levels can lead to the accumulation of sulfur-derived compounds or affect the metabolism of phenolic compounds, which can change the flavour, aroma, and texture of grapes and wine [40,48]. It was observed that residual sulfur on berries can lead to the formation of undesirable flavours, such as hydrogen sulfide (H$_2$S), during wine fermentation [48].

The flux distributions of the *V. vinifera* diel multi-tissue models were simulated to assess the effect of different sulfate concentrations on grapevine metabolism. Flux Variability

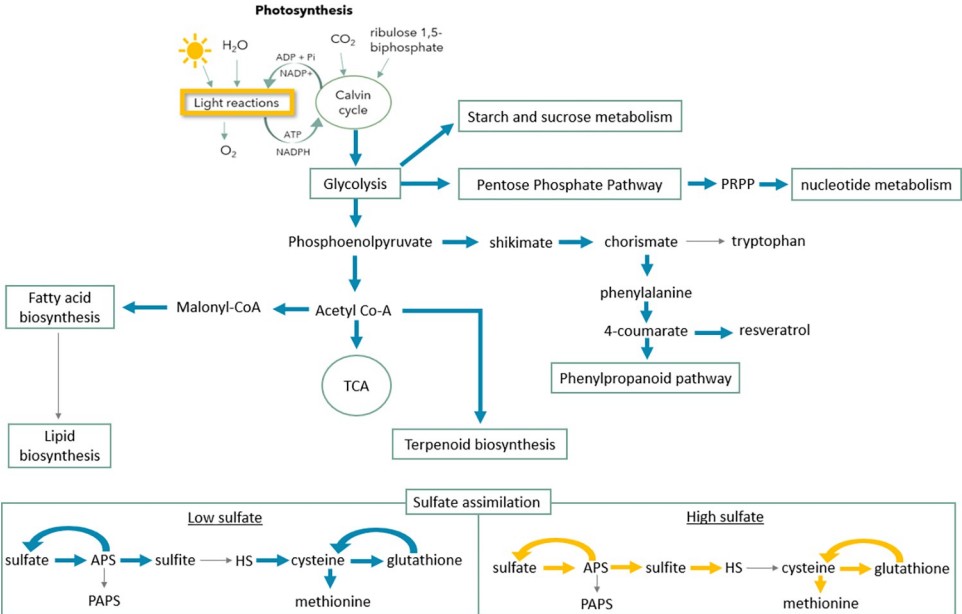

**Fig 4. Simplified schema of the main metabolic pathways in the model affected by varying sulfate levels.** Pathways with increased maximum flux under low sulfate conditions are highlighted with a thick blue arrow while pathways with increased flux under high sulfate conditions are highlighted with a thick yellow arrow. Pathways with decreased flux in both conditions are represented by a thin grey arrow.

Analysis (FVA) was used to get the possible range of reaction fluxes while keeping at least 80% of the maximum total biomass and fixing a photon uptake of 300 mmmol.gDW$^{-1}$.h$^{-1}$. Two different flux values for sulfate uptake in the light phase were tested, 0.01 and 10 mmmol.gDW$^{-1}$.h$^{-1}$. The choice of these values was arbitrary, but the goal was to have one value above and one below the unrestricted sulfate uptake flux (Table 4). Similar results were obtained for the multi-tissue models with green and mature berries. Thus, only the results for the green multi-tissue are described. The full results are available in S9 File and detailed in S1 File.

With high sulfate (10 mmmol.gDW$^{-1}$.h$^{-1}$), the maximum flux for biomass production decreased from 0.149 to 0.138 h$^{-1}$. Similarly, the production of all biomass components also decreased as well as the flux for primary and secondary metabolism (Fig 4).

As expected, the maximum fluxes of the reactions involved in sulfate assimilation and oxidation, and glutathione biosynthesis have increased. Surprisingly, in the model, the biosynthesis of cysteine and methionine decreased with high sulfate levels. During sulfate reduction, the reaction that produces H$_2$S has a higher maximum flux but the reaction that uses it to produce cysteine has a lower flux, which leads to a big increase in the flux of the H$_2$S exchange reaction (Fig 4). This could mean that when plants are exposed to high sulfur levels, they try to adapt to these conditions by adjusting their metabolism, leading to the accumulation of H$_2$S or other sulfur compounds that can alter the flavour and aroma of the grapes.

When plants are under a sulfate deficiency (uptake of 0.01 mmmol.gDW$^{-1}$.h$^{-1}$), the maximum production of biomass greatly decreased from 0.149 to 0.018 h$^{-1}$. Hence, the plant has an excess of carbon skeletons, which are not being used for protein biosynthesis and are available for the biosynthesis of secondary metabolites, increasing the available flux for these pathways. Therefore, there was an increase in the flux of primary pathways, such as sucrose and starch biosynthesis and degradation, gluconeogenesis, and glycolysis, as well as in the pathways responsible for producing secondary metabolites and plant hormones (Fig 4). For instance, the

maximum flux for resveratrol synthase reaction during the day increased from 0.41 to 1.75 mmmol.gDW$^{-1}$.h$^{-1}$. Thus, sulfur levels in the soil can greatly influence grapevine metabolism and affect the flavour and aroma of grapes by sulfide or sulfur-compound accumulation or changes in the phenolic content in grapes.

The same approach was applied to assess the effect of different nitrate concentrations in the *V. vinifera* model and similar patterns were observed. The full results are available in S10 File and detailed in S1 File.

## Machine learning and fluxomics

The potential of using ML to analyze fluxomics data generated by simulating the metabolic models was explored. As a significant number of samples is required to train good predictive ML models, the idea of applying ML to simulated fluxomics data is to study the relationships between variables and their impact on the output class. With this in mind, 73 context-specific GSMMs were created by integrating RNA-Seq data from grapes in different developmental stages. Simulated fluxomics data was then obtained from each GSMM as described in Materials and Methods. In this case, the output class to be predicted by the models is the grape developmental phase, green or mature.

First, data was preprocessed and explored using unsupervised methods, starting with t-SNE for data visualization [49] (S7 Fig in S1 File).

For the supervised analysis, five different ML model architectures were applied including logistic regression (LR), K-nearest neighbors (KNN), decision trees (DT), support vector machine (SVM), and random forests (RF). These ML models were evaluated using repeated stratified cross-validation with 10 folds and 10 repeats. The average evaluation results are shown in Table 6.

According to these results, the models are performing well in predicting the grape developmental stage with this fluxomics dataset. The model's performances across the different folds are robust, indicating that the models can learn meaningful patterns in the data and handle data variations well. Overall, RF obtained higher values for all metrics, while SVM presented the worst performance. Despite the results being good and the models being able to generalize correctly on the test set of each fold, the dataset is very small (73 samples), which is a common problem when working with omics data. Larger datasets are needed to create better predictive models and draw more conclusions from the data. Nevertheless, these models represent a good start for understanding which reactions contribute most to the model's prediction. Hence, SHAP values [50] were calculated for the two best models, RF and KNN, and the most contributing reactions are shown in Fig 5 for RF. The KNN results are described in S1 File.

Overall, when the reactions presented high flux capacity (FCa) values, they had negative SHAP values, leading the model to predict the green state, while with lower FCa they exhibited higher positive SHAP values, leading the model to predict the mature state. Hence, all the reactions identified here presented an average FCa value higher in the green than in the mature grapes.

For the RF model, RXN0-882 in the chloroplast is the reaction that most contributes to the predictions. High fluxes of this reaction have negative SHAP values (around -0.15), classifying

**Table 6. Evaluation results of the ML models for the metrics balanced accuracy, precision, recall, and F1 score.**

|  | LR | KNN | DT | SVM | RF |
|---|---|---|---|---|---|
| **BALANCED ACCURACY** | 0.96 | 0.95 | 0.96 | 0.93 | 0.97 |
| **PRECISION** | 0.97 | 0.96 | 0.98 | 0.96 | 0.98 |
| **RECALL** | 0.96 | 0.98 | 0.95 | 0.94 | 0.97 |
| **F1 SCORE** | 0.96 | 0.97 | 0.96 | 0.94 | 0.97 |

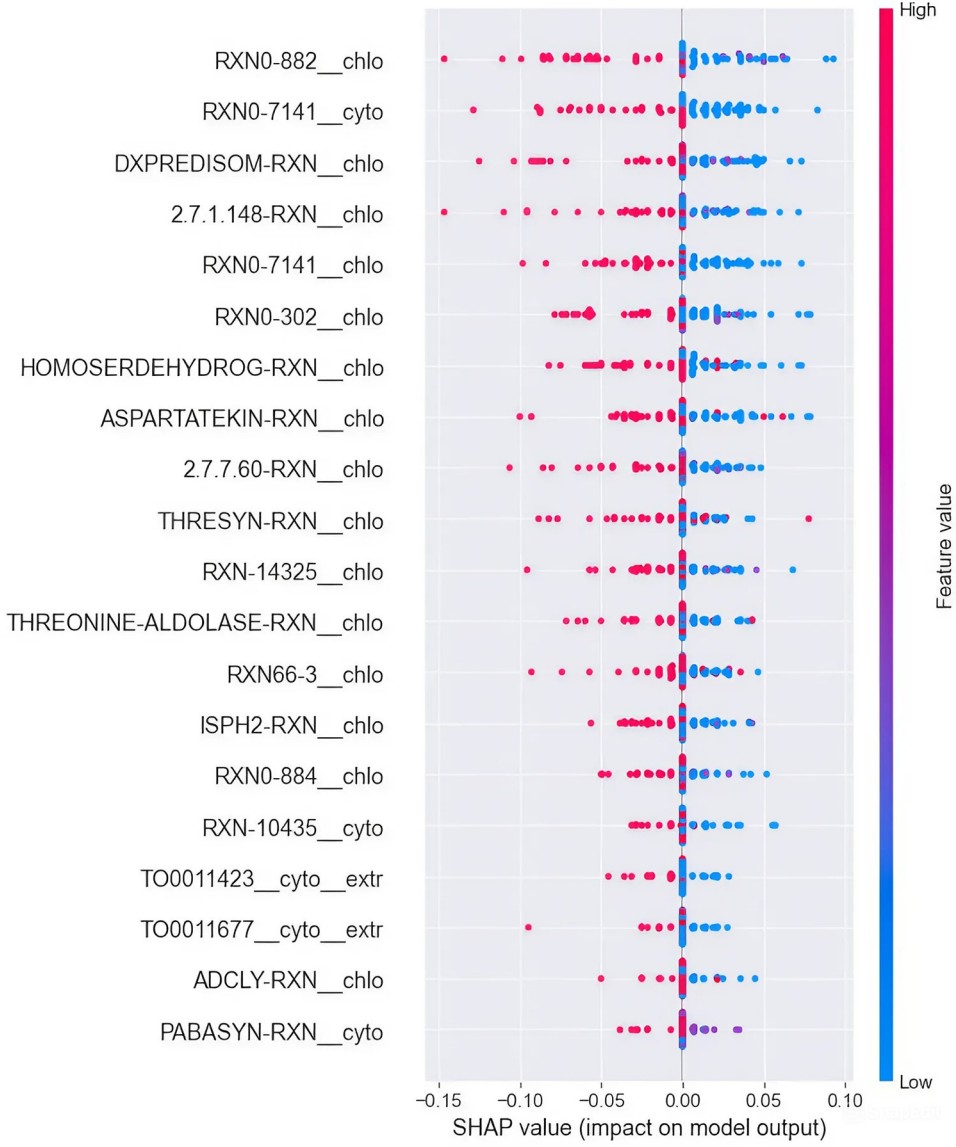

**Fig 5. Beeswarm plot of SHAP values for the reactions that contribute most to RF's predictions.** Features are ordered from higher to lower effects on the predictions. The dots represent a single observation, and the color indicates if the observation has a higher (pink) or a lower (blue) feature value compared to the other observations.

the samples as green, while lower fluxes have positive SHAP values (close to 0.10), classifying the samples as mature. There are some exceptions to this trend, such as the chloroplastic THREONINE-ALDOLASE-RXN and THRESYN-RXN reactions that show positive SHAP values when presenting high flux, indicating that samples classified as mature by the models can also have high fluxes in these reactions.

Of the 20 reactions identified for each model, 10 have a high impact on both models, indicating that the results are reliable and robust and that these features are important for predicting the output. Most of these reactions are involved in the methylerythritol phosphate (MEP) pathway, which is responsible for the biosynthesis of the terpenoid precursors (S5 Fig in S1 File), threonine degradation into glycine, and the transport of glycerides. The remaining

reactions identified only with the RF model are involved in the biosynthesis of nucleotides, 4-aminobenzoate, and threonine.

The accumulation of terpenoids in grapes typically starts before veraison, which can explain why the reactions associated with the biosynthesis of terpenoid precursors had higher FCa in the green state. However, terpenoid biosynthesis intensifies after veraison, which is not observed in the fluxes of these reactions. Fasoli et al. [51] have also identified terpene metabolism as a negative biomarker for the onset of ripening. In addition, the abscisic acid (ABA) signalling is increased at veraison, and ABA is derived from carotenoids, whose biosynthesis starts with the MEP pathway. Thus, there is strong evidence that genes or reactions from the MEP pathway could be used as biomarkers for the onset of ripening.

In the green phase, as grapes are rapidly growing, the metabolism of amino acids, nucleotides, and lipids is expected to be more active than in the mature phase. 4-aminobenzoate is a precursor for the biosynthesis of tetrahydrofolates, which are involved in several processes like photorespiration, amino acid metabolism, and protein biosynthesis. These pathways are also expected to be more active in the green phase. This fact may explain why the reactions related to these pathways are important for the model's predictions. However, it is not clear why threonine metabolism is more important for the model than the metabolism of the other amino acids. Nevertheless, the models presented good predictions, associating high fluxes of these reactions to predict the green state and low fluxes to predict the mature state.

## Materials and methods

### Metabolic data source

The metabolic information of PlantCyc 14.0 [29] and MetaCyc 26.1 [30] databases was saved and organized in a repository named *iplants*, using Neo4j [52] and MongoDB [53] database management systems. These NoSQL databases do not store data in relational tables, having instead a more flexible schema. Neo4j uses a graph structure, while MongoDB uses a document structure to represent data. The implementation and management of the databases were performed with Python 3.8, using the Neomodel package, an object graph mapper for Neo4j, and Mongoengine, an object document mapper for MongoDB. The Neo4j database saved the connections between metabolites, reactions, enzymes, genes, pathways, organisms, and models, while MongoDB stored metadata for all Neo4j entities. UniProt [31] data for enzymes were also collected when available, and added to MongoDB, including protein function, localization, sequence, and annotation status.

In addition, nine plant metabolic models of different species were integrated into the repository, which involved matching the *iplants* entry identifier with the identifier in the models and connecting the model object with the associated objects in the database.

### Model reconstruction

The reconstruction of the GSMM of *V. vinifera* was based on the PN40024.v4 genome and the PN40024.v4.1 annotation version, which includes 35922 genes and 41160 proteins [28]. All the main steps and analyses were performed using Python 3.8 and *COBRApy* version 0.25 [54] and are schematized in S2 Fig in S1 File.

The reconstruction started with genome annotation which was based on DIAMOND [36] similarity searches of *V. vinifera* proteins against the protein sequences in *iplants*, to assign enzymatic functions and associate reactions to *V. vinifera* proteins. Using only the *iplants* database to perform the annotation facilitated the identification of the reactions to include in the model. DIAMOND searches were performed against the SwissProt database to determine the completeness of important enzyme annotations in *iplants*. In the course of these searches, it was

found that more than 17,000 *V. vinifera* proteins had a match in SwissProt. However, most proteins were involved in other processes like transcription regulation, protein phosphorylation, and nuclear transport, which are not relevant to be included in the metabolic model.

Based on the *iplants*' annotation, the reactions linked to the identified enzymes and spontaneous reactions were assembled to create a draft metabolic network. WolfPsort [55] and LocTree3 [56] were used to predict the subcellular location of the proteins. However, due to contradictory outcomes from these tools, manual curation of the results was deemed necessary. For instance, certain enzymes catalyzing the reactions of the sphingolipid biosynthesis, like serine C-palmitoyltransferase and dihydroceramide fatty acyl 2-hydroxylase reactions, were predicted to be on the endoplasmic reticulum by Loctree3 and in the chloroplast or cytosol by WolfPSort. In this case, the annotations collected from UniProt were considered and the location of these enzymes was defined to be the endoplasmic reticulum. The thylakoidal and mitochondrial intermembrane compartments were added manually for the photosynthesis and oxidative phosphorylation reactions, respectively.

Transport reactions were automatically identified using TranSyT [57] and added to the model. Additional transporters were manually included in the model when required. All reactions were validated for mass and charge balance.

## Biomass composition

The definition of tissue-specific biomass compositions is crucial to obtaining good models capable of simulating the specific metabolism of each tissue. Biomass is composed of macromolecules labelled "e-metabolites", which are required for cell growth, including RNA, DNA, proteins, carbohydrates, lipids, co-factors, and cell wall components. Ideally, experimental data should be used to define biomass composition for different tissues. However, as these data are not available for *V. vinifera*, the biomass content was estimated based on previously published plant GSMMs and insights from the literature. Specifically, the biomass formulation for leaf, stem, and green berry was based on the models of *A. thaliana* [6,10], *S. lycopersicum* [32], and *Q. suber* [38], while the biomass of the mature berry was adjusted according to the literature [40] and the metabolomics data available for the same samples used to obtain the RNA-Seq data [51]. Details of the biomass composition are available in S4 File.

The monomer contents for the production of DNA, RNA, and proteins were calculated from the genome sequence using the biomass tool [58], available in *merlin* [59]. The reactions for the production of the cell wall, carbohydrates, fatty acids, lipids, and co-factors were also adapted from *A. thaliana* and *Q. suber* models. The e-Cofactor metabolite includes several universal cofactors, such as NAD(H), and vitamins, and in the case of leaf, it also includes pigments, such as chlorophylls. The content of carbohydrates was adapted to reflect the grape composition described in the literature. For instance, tartaric acid was added to the model as it was described as the main organic acid found in grapes and it is not present in any other plant model [40]. For mature berries, the content of sugars and organic acids was adjusted to reflect the changes in berry composition during maturation. In addition, some secondary metabolites were added to the carbohydrate reaction of the mature berry based on metabolomics data [51]. These metabolites were found in mature grapes and mainly included anthocyanins, such as malvidin 3-glucoside, peonidin 3-O-glucoside, and petunidin-3-O-glucoside.

As no information regarding the energetic requirements for *V. vinifera* was found in the literature, it was retrieved from the model of *S. lycopersicum* [32]. The growth-associated maintenance energy requirement was included in the biomass equation and defined as 53.26 mmmol. $gDW^{-1}$. The nongrowth-associated maintenance energy requirement was defined in an ATP hydrolysis reaction with a mandatory flux of 2 mmmol.$gDW^{-1}$.h.

**Manual validation.**    Manual curation was an essential step during the reconstruction process. Literature and biological databases, such as Kyoto Encyclopedia of Genes and Genomes (KEGG) [60], National Center for Biotechnology Information (NCBI) [61], BRaunschweig Enzyme Database (BRENDA) [62] and UniProt [31], were consulted to retrieve additional information about specific reactions, enzymes, or pathways.

Validating the model required verifying its ability to produce biomass. *BioISO* [63] was used to accomplish this goal, as it verifies the production of each biomass substrate to see which ones are missing. Then, the gaps in the model were analyzed and filled in when necessary to produce essential metabolites. Also, it was assured that growth did not occur without photons and carbon sources, and no futile cycles were present. Then, dead-end metabolites and blocked reactions were identified, and each blocked reaction was analyzed and fixed by resorting to other databases. When no information was available, the blocked reactions were kept in the model.

Finally, the model's capability to accurately simulate key metabolic processes like photosynthesis, photorespiration, and respiration was confirmed. This was achieved through the application of diverse methods, including Flux Balance Analysis (FBA), parsimonious FBA (pFBA), and Flux Variability Analysis (FVA). FBA [64], which uses linear programming to calculate an optimal flux distribution for a given objective function, has been extensively used to simulate GSMMs. However, multiple optimal solutions usually exist in the solution space for a given objective. Hence, pFBA has emerged as a novel approach. It refines the traditional FBA by selecting a flux distribution from the FBA optimal space that minimizes the total sum of fluxes [42]. Likewise, FVA [65] is used to determine the span of flux variability of GSMMs in simulations by calculating the minimum and maximum flux of each reaction for a defined set of constraints.

## Tissue-specific models

**Omics data.**    Available RNA-Seq data from different tissues were used to create specific GSMMs for stem, leaf, and berry. Due to the absence of a single study providing RNA-Seq data for all three tissues, we resorted to using two distinct studies to gather the necessary information. Leaf and stem data were retrieved from the healthy samples of *V. vinifera* Cabernet Sauvignon cultivar in the study of Massonnet et al. [66] (Gene Expression Omnibus (GEO) accession: GSE97900). RNA-Seq data for the C. Sauvignon berries was obtained from the study of Fasoli et al. [51] (GEO accession: GSE98923), which included berry samples in different developmental stages and associated metabolomics data. The time-point metadata for these samples was grouped into two developmental stages, green and mature, to create a metabolic model for each state. Samples until time point 4 were considered to be in the green state, while samples after time point 4 were considered to be mature. Although sample time 4 was collected 7 days after veraison, it was still considered to be in the green phase to reduce the class imbalance.

These datasets were retrieved from GRape Expression ATlas (GREAT) [39]. GREAT is a gene expression atlas for grapevine that integrates all public RNA-Seq experiments, allowing the analysis and visualization of the data. The RNA-Seq data were already normalized in transcripts per million (TPM) and the reads were mapped to the new genome (PN40024.v4).

A dataset with all collected data and respective metadata was built, and a log2 transformation was applied to the gene expression values. Finally, the gene identifiers in the datasets were mapped to the protein identifiers present in the model to allow for omics integration.

**Models.**    Tissue-specific models for stem, leaf, green berry, and mature berry were reconstructed using the FASTCORE algorithm [41] implemented in the *Troppo* package [67]. This

algorithm identifies the reactions that should be removed or kept in the model, based on the expression levels of the genes associated with each reaction through GPR rules, resulting in models with different reaction content.

The reconstructed generic model and the omics dataset were used as input for this algorithm, and the local T2 thresholding strategy [68] was applied to preprocess the omics data before integrating it into the model. In this strategy, two global thresholds, upper and lower, are defined, and genes whose expression is below the lower threshold are considered to be inactive, while genes whose expression is above the upper threshold are considered to be active. Genes with expression levels between these two thresholds may be active or not and further analysis is performed by comparing the expression values to a local threshold. This is a gene-specific threshold that accounts for the expression levels of each gene over all samples, while the global thresholds have the same value for all genes. This strategy was employed as it was described to obtain better results [68].

In this work, we selected the percentiles 25 and 75 for the global lower and upper thresholds, respectively, and the percentile 50 (median) for the local threshold. The pseudo-reaction representing the drain of macromolecules required to create a new unit of biomass was included in the set of protected reactions of the algorithm so that all tissue-specific models would be able to produce biomass.

**Phenotype predictions.** Phenotype predictions of the tissue-specific models were performed by pFBA, using two different strategies as applied in other plant models [6,38] and based on *A. thaliana* experimental measures [69]. The first consisted of fixing the biomass growth rate to $0.11 h^{-1}$ and defining the minimization of the photon/sucrose uptake as the objective function. The second strategy defined the biomass growth rate as the objective function and fixed the photon uptake to 100 $mmmol.gDW^{-1}.h^{-1}$ for photosynthesis and photorespiration and the sucrose uptake to 1 $mmol.gDW^{-1}.h^{-1}$ for respiration. Photorespiration was simulated by constraining the carboxylation (RIBULOSE-BISPHOSPHATE--CARBOXYLASE-RXN) and oxygenation (RXN-961) reactions by Rubisco with a flux ratio of 3:1 [6,38].

**Differential flux analysis.** Differential flux analysis between the created tissue models was performed using the approach of [70]. In this approach, sample fluxes for the tissue models were generated using Artificial Co-ordinate Hit and Run (ACHR) sampler [46] from *CobraPy*, with a thinning factor of 100 and a sample size of 10000 for each model. Pairwise Kolmogorov-Smirnov tests were used to compare the flux distribution of the distinct tissue models. The flux change (FC) of each reaction between the two models was also calculated as shown in Eq 1, where $\bar{S}_{model1}$ and $\bar{S}_{model2}$ represent the mean of the flux distributions for a reaction in *model1* and *model2*, respectively. Reactions with an absolute value less than 0.82 (equivalent to a 10-fold change in flux) were considered insignificant.

$$FC = \frac{\bar{S}_{model1} - \bar{S}_{model2}}{|\bar{S}_{model1} + \bar{S}_{model2}|}$$

(1)

For reactions that are absent in a model, their flux is assumed to be zero in that model, and bootstrapping is used to estimate the 95% confidence interval of their fluxes. If zero is outside the interval, the reactions are considered to have differential flux in the two models. In addition, the p-values of altered reactions were adjusted by a Benjamini-Hochberg correction, with a significance level of 0.05. The differential pathways between models were obtained using hypergeometric enrichment tests that select the pathways that are over-represented due to the higher number of altered reactions and not by chance.

### Diel multi-tissue models

**Models.**   Multi-tissue models were created by joining the tissue-specific models and connecting them by two common pools, one between stem and leaf, and the other between stem and berry, based on a previous approach [10]. Transport reactions between tissues and common pools were manually added when required. Given that a model for the root was not developed, we assumed that the uptake of minerals occurs in the stem. Exchanges of water, oxygen, and carbon dioxide were allowed in all tissues, and light absorption was allowed only in the leaf model. Two multi-tissue models were created, one with the berry in the green phase and the other with the mature berry.

Additionally, diel models were created to account for light and dark phases. All reactions and metabolites were duplicated for each phase, and new reactions were added to allow the exchange of some metabolites between the two phases. These are called storage metabolites and include the 20 amino acids, nitrate, citrate, malate, glucose, sucrose, fructose, and starch, which can be produced in one phase and used in the other, as previously described [9].

**Phenotype predictions.**   Phenotype predictions using multi-tissue diel models were also performed with pFBA using the second strategy mentioned above for photorespiration conditions, but with a photon uptake of 300 mmmol.gDW$^{-1}$.h$^{-1}$ as flux values were very low with 100 mmmol.gDW$^{-1}$.h$^{-1}$. As in other plant diel models [9,11,38], the nitrate uptake was constrained to a ratio of 3:2 in the light and dark cycle.

### Machine learning and fluxomics

**Data.**   All samples from the RNA-Seq dataset from Fasoli et al. [51] were used to create simulated fluxomics data for grapes in the green and mature state, by using the iMS7199 model and the aforementioned phenotype prediction approaches to reach flux distributions. Firstly, the mean expression value of all replicates was calculated to represent each biological sample, resulting in a dataset with 73 samples, 55% from Cabernet Sauvignon and 45% from Pinot Noir, and the log2 expressions of the 6018 genes in the model. As performed before, these samples were discretized into two developmental stages, green and mature, which represent the output class to be later predicted by the ML models. Then, the final RNA-Seq dataset was integrated with the generic *V. vinifera* model to create GSMMs representing each sample in the dataset using the FASTCORE algorithm as described before. In total, 73 context-specific models were created, one per sample. The resulting sample-specific GSMMs were simulated using FVA and the flux capacity (FCa) of each reaction was calculated by subtracting the maximum and minimum flux obtained for each reaction while keeping 80% of the maximum biomass value (Eq 2). Before running FVA, all reactions were made irreversible to facilitate the interpretation of results. The reactions absent in the models were considered to have a capacity of 0.

$$FCa(r) = Flux_{max}(r) - Flux_{min}(r) \tag{2}$$

**Models.**   The analysis of fluxomics with ML was performed in Python 3.11 with Scikit-learn 1.2.2. For unsupervised analysis, the dataset was filtered to remove the reactions with the same FCa across all samples using *VarianceThreshold* and scaled by *StandardScaler*. t-SNE was applied to visualize the distribution of the data. For the supervised analysis, the original dataset was divided into train and test sets by cross-validation with 10 folds and repeated 10 times, using the *RepeatedStratifiedKfold* function. In each iteration, feature selection was performed using *VarianceThreshold* and, as the number of features was still high, the *SelectKBest* function was used to select the 500 most relevant features based on ANOVA F-values. In addition, the

resulting dataset was also scaled by *StandardScaler*. Then, an ML model fitted the train data and predicted the output classes for the test set. Five different ML models were tested including logistic regression, K-nearest neighbors, decision trees, support vector machine, and random forests. These were evaluated by different metrics, such as recall (Eq 3), precision (Eq 4), balanced accuracy (Eq 5), and F1 score (Eq 6), which were averaged across all train-test splits.

$$recall = \frac{TP}{TP + FN} \tag{3}$$

$$precision = \frac{TP}{TP + FP} \tag{4}$$

$$balanced\ accuracy = 0.5 * \left( recall + \frac{TN}{TN + FP} \right) \tag{5}$$

$$F1\ score = \frac{2 * TP}{TP + 0.5(FP + FN)} \tag{6}$$

The importance of each feature in the prediction of the output was analyzed by calculating the SHAP values for each classifier. SHAP values (SHAPley Additive exPlanations) are defined based on the contribution of each feature to the prediction of each sample and are used to increase the interpretability of ML models. Larger absolute SHAP values have a larger effect on the prediction [50]. The SHAP values for each fold of the repeated cross-validation were calculated, and the average SHAP values for each sample were calculated to give a more stable representation of the feature contributions.

## Conclusion

In this work, diel multi-tissue models of *V. vinifera* were reconstructed by integrating tissue-specific omics data for leaf, stem and grapes. The iMS7199 model is available in the BioModels database (https://www.ebi.ac.uk/biomodels/MODEL2408120001), together with the FROG report that shows the model's robustness and reproducibility. Likewise, the tissue-specific and diel models have also been published in the biomodels database https://www.ebi.ac.uk/biomodels/MODEL2408160001). These models confirm the effect of different sulfate and nitrate concentrations on the metabolic content of the grapes. In addition, the flux distributions obtained from grape GSMMs were analysed by ML models and the most contributing reactions are involved in MEP and threonine biosynthesis and could represent putative biomarkers of the grape development stage.

The main limitation of these GSMMs is the lack of experimental data to define pseudo-reactions that represent the drain of macromolecules to create a new unit of biomass, and to validate the phenotype predictions. Nevertheless, these pseudo-reactions can be easily updated when actual data on the biomass composition from *V. vinifera* become available. In addition, the absence of significant differences between the green and mature grape models may be associated with the use of time-course omics data, which can hinder the identification of state-specific metabolic patterns. Hence, these models could obtain better results when applied to contrasting conditions. For instance, in future work, the model can be integrated with other datasets to investigate condition-specific metabolic phenotypes, such as responses to environmental stresses, like drought, or to understand disease mechanisms and developmentally regulated pathways, thereby offering a more comprehensive understanding of the plant's metabolism. Additionally, applying ML to analyse the fluxomics data predicted by the GSMMs

can be greatly improved when larger omics datasets become available, allowing for more representative results.

## Supporting information

**S1 File. Technical and additional details about the methods and results.**
(PDF)

**S2 File. List of the attributes for all objects in MongoDB.**
(XLSX)

**S3 File. Full comparisons between the metabolic content of iMS7199 and other plant models.**
(XLSX)

**S4 File. Details on the definition of biomass compositions.**
(XLSX)

**S5 File. Pathway content by tissue.**
(XLSX)

**S6 File. pFBA phenotype predictions of each tissue GSMM.**
(XLSX)

**S7 File. Full differential flux analysis results.**
(XLSX)

**S8 File. pFBA phenotype predictions of the diel GSMMs.**
(XLSX)

**S9 File. Effects of varying sulfate levels on the phenotype predictions of the diel GSMMs.**
(XLSX)

**S10 File. Effects of varying nitrate levels on the phenotype predictions of the diel GSMMs.**
(XLSX)

## Author Contributions

**Conceptualization:** Miguel Rocha, Oscar Dias.

**Data curation:** Marta Sampaio.

**Formal analysis:** Marta Sampaio.

**Investigation:** Marta Sampaio.

**Methodology:** Marta Sampaio.

**Project administration:** Miguel Rocha, Oscar Dias.

**Resources:** Miguel Rocha, Oscar Dias.

**Software:** Marta Sampaio.

**Supervision:** Miguel Rocha, Oscar Dias.

**Validation:** Marta Sampaio.

**Visualization:** Marta Sampaio.

**Writing – original draft:** Marta Sampaio.

**Writing – review & editing:** Miguel Rocha, Oscar Dias.

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
