## [Decision Letter · Decision Letter 0]

17 Apr 2024

Dear Ms Sampaio,

Thank you very much for submitting your manuscript "A diel multi-tissue genome-scale metabolic model of Vitis vinifera" for consideration at PLOS Computational Biology.

As with all papers reviewed by the journal, your manuscript was reviewed by members of the editorial board and by several independent reviewers. In light of the reviews (below this email), we would like to invite the resubmission of a significantly-revised version that takes into account the reviewers' comments. Please pay particular attention to the comments relating to the clarity as well as structure of the manuscript and showcasing the use of your model.

We cannot make any decision about publication until we have seen the revised manuscript and your response to the reviewers' comments. Your revised manuscript is also likely to be sent to reviewers for further evaluation.

Sincerely,

Christoph Kaleta

Academic Editor

PLOS Computational Biology

Pedro Mendes

Section Editor

PLOS Computational Biology

Reviewer's Responses to Questions

**Comments to the Authors:**

Reviewer #1: In this manuscript, Sampaio et al. present the reconstruction of the genome-scale metabolic model of the plant Vitis vinifera. Their aim is to reconstruct a diel multi-tissue model capable of reproducing the main metabolic landscape of this organism. I appreciate the effort that the authors made in reconstructing the model and in including many specific pathways that were not covered in previous reconstructions. Also, they made extensive use of manual curation, which is good. I loaded the model and ran a few basic simulations and everything worked correctly. Surely, the model will be a valuable contribution to those working in the field of plant systems biology and/or functional genomics. The authors show that their model is more comprehensive than other currently available plant models and, as such, it is a nice addition to the actual literature. Apart from this strong effort in the reconstruction phase, the manuscript lacks a bit in originality and does not provide many new and/or innovative aspects of plant metabolism.

Also, the reading of the manuscript is quite tough. The text is too long and quite hard to follow, especially in the first part. I would encourage the authors to present most of the text and figures of pages 10 to 14 (surely figures 3 and 4) as additional material. This would make the manuscript easier to read and allow the authors to focus most on the use of the model rather than its assembly. There might be other sections that could be presented as additional material in the manuscript so my suggestion is to go through the manuscript once more and try to reduce the text (and maybe figures) to improve the overall readability.

I also think that the authors could use the model in a more “propositive” and innovative way to explore some key issues of plant metabolism. A few examples that the authors might want to take into consideration.

Could the grape models at different stages of maturations be used to infer which are the key pathways that affect (or maybe start) the process of grapes rotting?

Could the same models be used to identify targets that could be exploited to improve and/or modify the metabolic content of grapes and, as a consequence, the taste of wine?

These are just a few examples. The authors may find other interesting and alternative points to address with their reconstruction.

Specific points:

Line 150 (and elsewhere?): (Fig. 1Error! Reference source not found.). Please correct.

Lines 154 - 158: Do these differences in the number of reactions/genes/metabolites reflect the evolutionary differences between the species or simply reconstruction efforts?

Line 299 (and other): “The leaf tissue was simulated for all processes”. I think here the authors meant “The flux distribution in the leaf was simulated …”. I advise the use of this form throughout the text.

Line 301: “Photosynthesis can also occur in green berries, but not at significant levels”. A reference is needed here.

Line 336: -> “Among models”

Line 506: “When plants are under a sulfate deficiency, the production of biomass and all its

components also decreases.” Sulfate likely participates in many reactions of the model and is probably included in the biomass reaction (coenzymes?), thus it is not surprising that reducing its uptake from the environment lower biomass is produced and fluxes are generally low.

Line 524: Machine Learning and Fluxomics paragraph. In this section it is not clear that fluxomics data is coming from constraint-based simulations. Please make it clear early in the paragraph that this is the case.

Reviewer #2: The manuscript “A diel multi-tissue genome-scale metabolic model of Vitis vinifera” by Sampaio et al. describes (i)the iPlants repository, a collection of relevant data for reconstruction of metabolic data and also nine published and publicly available plant metabolic models; (ii)the reconstructed GSMM of V. vinifera and compared the reaction contents of their model with other seven plant metabolic models; (iii) pathway distributions of the reactions present in their model but not in other models; (iv) integration of RNA-seq data with the GSMM to get tissue specific models; (v) diel-multi tissue (leaf, stem, berry) model; (vi) how the machine learning method is used to analyze flux data of different content-specific GSMMs. Further, some simulation results were shown to match with known biochemical active pathways in specific condition.

However, I cannot recommend to accept the manuscript in its present form due to following reasons:

1. The authors claim in their abstract that “advances have been made, allowing the integration of omics datasets with GSMMs”. It is not clear in the manuscript what is the specific advancement achievement in this paper. Integration of omics data as well as multi-tissue modeling was reported earlier. One such examples is

“Reconstruction of Arabidopsis metabolic network models accounting for subcellular compartmentalization and tissue-specificity. https://doi.org/10.1073/pnas.1100358109”

Nadine Töpfer, Camila Caldana, Sergio Grimbs, Lothar Willmitzer, Alisdair R. Fernie, Zoran Nikoloski, Integration of Genome-Scale Modeling and Transcript Profiling Reveals Metabolic Pathways Underlying Light and Temperature Acclimation in Arabidopsis , The Plant Cell, Volume 25, Issue 4, April 2013, Pages 1197–1211, https://doi.org/10.1105/tpc.112.108852

Interestingly, these types of breakthrough relevant works have not been cited/discussed in this current manuscript. Thus, it is incomplete.

2. The authors state “Furthermore, to capture the dynamic changes in grape metabolism, we created two separate models representing the grape in both its green and mature states.” I have not found any section that clearly explains/describes the dynamic changes they have been able to capture.

3. The manuscript is very badly represented. For example, the limitations like Line 273: “The biomass of leaf and green berry was considered to be the same.” should be discussed separately in a subsection. Whether they have used any maintenance cost in the model is not clear. Many figures (Figures 2,3,4) may be removed to supplementary.

Reviewer #3: General remarks and main concerns:

In this manuscript, the authors reconstructed a generic genome-scale metabolic model of grapevine metabolism. They explore the metabolism with a model combining tissue-specific models based on omics data of stem, leaf, and berry, and day-night modes. Fluxes were analysed by ML. The manuscript is written relatively clearly and the work carried out is substantial in terms of bioinformatics tools used for the reconstruction of the model, but the flux analyses and the ML methods used (t-SNE) are poorly justified and not always relevant.

The biggest concern is that the reconstructed model is proven to account for physiology by taking very coarse and global data, such as for example the same biomass composition for the leaf and the young berry, a protein content of 60% in the mature berry (huge and higher than young berries) or the same growth rate for all tissues. This information cannot properly account for flux predictions in line with real phenotypes. Even if there are no data for grapevines, it would be preferable to take tissue-specific data from another plant. This crude parameterization can largely explain the lack of phenotype difference in the predicted fluxes. Also, the multi-tissue model is reduced to 2 stages of berry development, so it is inappropriate to talk about metabolic changes during development. And beyond the separation of tissues and the 2 stages, on the basis of differential flows, the model does not provide relevant physiological conclusions. Why are only differential flows looked at? Also, the ML method used, t-SNE, is not justified, what does it offer compared to a more traditional PCA? why not look for the variables explaining the separations?

As for the day-night model, whether for young or ripe berries, the phenotype predictions (table 4) are very similar; It’s surprising and disappointing. Also, an equivalent flux of biomass is impossible because the mature berry is no longer growing, unlike the young berry. Finally, the last part using ML on fluxomics data is complex and does not clearly highlight the underlying results obtained whether with t-SNE or with the predictions. For the interpretation part of the variables underlying the predictions, why is it mentioned that 10 of the 20 reactions have a strong impact? Should the same reactions have a strong impact for all 4 models? Why did the authors have to calculate flow capacity, in addition to flux estimates?.

Finally, there are many figures (12) and some are not very relevant for this work (figs 3 and 4) or not very useful, like fig 5. There are also many tables (actually 6). Some of them are difficult to understand such as Table 5 with the names of the reactions are not explicit.

Minor concerns

L480 : why is the FVA done here and not before?

L 788 : the nitrate uptake was constrained to a ratio of 3:2, in terms of what?

l 271, l 398 and later : typography problem to cite tables

Legend fig 5 is not “RNA-Seq data for all tissues” but the percentage of RNA-Seq analyzes for all tissues

Legend fig 6 is not “Biomass composition “ but percentage (?) of the main biomass compound in the biomass

**Have the authors made all data and (if applicable) computational code underlying the findings in their manuscript fully available?**

Reviewer #1: Yes

Reviewer #2: Yes

Reviewer #3: Yes

PLOS authors have the option to publish the peer review history of their article (what does this mean?). If published, this will include your full peer review and any attached files.

Reviewer #1: **Yes: **Marco Fondi

Reviewer #2: No

Reviewer #3: No
---

## [Decision Letter · Decision Letter 1]

23 Jul 2024

Dear Ms Sampaio,

Thank you very much for submitting your manuscript "A diel multi-tissue genome-scale metabolic model of Vitis vinifera" for consideration at PLOS Computational Biology. As with all papers reviewed by the journal, your manuscript was reviewed by members of the editorial board and by several independent reviewers. The reviewers appreciated the attention to an important topic. Based on the reviews, we are likely to accept this manuscript for publication, providing that you modify the manuscript according to the review recommendations. Please pay in particular attention to the comments related to applicability but also limitations of your study.

Sincerely,

Christoph Kaleta

Section Editor

PLOS Computational Biology

Pedro Mendes

Section Editor

PLOS Computational Biology

Reviewer's Responses to Questions

**Comments to the Authors:**

Reviewer #1: I acknowledge the effort that the authors made in reconstructing such a large and complex model. I think it will be a valuable resource in the next years. At the same time, I feel like the authors could/should have put more effort in simplifying and shortening the text. The revised version has been reduced by approx. 1 and a half page. Also, I am not entirely convinced by the fact that there are no sufficient -omics data to interrogate the model further and provide more insights into the metabolism of this plant in disparate conditions. V. vinifera is one of the best studied plants and there are many interesting RNAseq datasets in the databases. I am not saying that the authors should add some additional analyses here but it would be nice to have a sentence or two on the possible use of the reconstruction to investigate some specific metabolic phenotypes of V. vinifera in the future.

Reviewer #2: The authors have tried to address the queries and suggestions of myself and other reviewers. However, I think that a different subsection describing the limitations of the present study and how these limitations do not affect the conclusion of the present work is very much needed. It is even clear from the author's response to reviewers that there are several limitations (some limitations due to lack of the data while some others are due to lack of techniques). This subsection will help the reader to judge the predictions of this model and at the same time it will help other researchers to make any advancement of the current study.

The authors may be requested (extending the queries of the third reviewer) to show how the traditional PCA like methods fail to reach the same conclusion what the authors have obtained using ML method.

**Have the authors made all data and (if applicable) computational code underlying the findings in their manuscript fully available?**

Reviewer #1: Yes

Reviewer #2: Yes

PLOS authors have the option to publish the peer review history of their article (what does this mean?). If published, this will include your full peer review and any attached files.

Reviewer #1: No

Reviewer #2: No

Figure Files:

Data Requirements:

Reproducibility:

References:

---

## [Decision Letter · Decision Letter 2]

23 Sep 2024

Dear Ms Sampaio,

We are pleased to inform you that your manuscript 'A diel multi-tissue genome-scale metabolic model of Vitis vinifera' has been provisionally accepted for publication in PLOS Computational Biology.

Best regards,

Christoph Kaleta

Section Editor

PLOS Computational Biology

Christoph Kaleta

Section Editor

PLOS Computational Biology

Reviewer's Responses to Questions

**Comments to the Authors:**

Reviewer #1: All my observations have been addressed. Thanks and congrats.

Reviewer #2: The authors have included most of the suggestions from the reviewers and have discussed the limitations of the work.

**Have the authors made all data and (if applicable) computational code underlying the findings in their manuscript fully available?**

Reviewer #1: Yes

Reviewer #2: None

PLOS authors have the option to publish the peer review history of their article (what does this mean?). If published, this will include your full peer review and any attached files.

Reviewer #1: No

Reviewer #2: **Yes: **Sudip Kundu

---

## [Editor Report · Acceptance letter]

27 Sep 2024

PCOMPBIOL-D-24-00340R2 

A diel multi-tissue genome-scale metabolic model of Vitis vinifera

Dear Dr Sampaio,

I am pleased to inform you that your manuscript has been formally accepted for publication in PLOS Computational Biology. Your manuscript is now with our production department and you will be notified of the publication date in due course.

With kind regards,

Anita Estes
